

# Semi-field evaluation of human landing catches *versus* human double net trap for estimating human biting rate of *Anopheles minimus* and *Anopheles harrisoni* in Thailand

Chanly Yan[1], Jeffrey Hii[2], Ratchadawan Ngoen-Klan[1], Manop Saeung[1] and Theeraphap Chareonviriyaphap[1,3]

[1] Department of Entomology, Faculty of Agriculture, Kasetsart University, Bangkok, Thailand
[2] College of Public Health, Medical and Veterinary Sciences, James Cook University of North Queensland, North Queensland, Australia
[3] Royal Society of Thailand, Bangkok, Thailand

Corresponding author
Theeraphap Chareonviriyaphap, faasthc@ku.ac.th

## ABSTRACT

**Background**. Whilst the human landing catch (HLC) technique is considered the 'gold standard' for estimating human-biting rates, it is labor-intensive and fraught with potential risk of exposure to infectious mosquito bites. This study evaluated the feasibility and performance of an alternative method, the human double net trap (HDNT) relative to HLC for monitoring host-seeking malaria vectors of the *Anopheles minimus* complex in a semi-field system (SFS).

**Methods**. HDNT and HLC were positioned in two rooms, 30 m apart at both ends of the SFS. Two human volunteers were rotated between both traps and collected released mosquitoes ($n = 100$) from 6:00 pm till 6:00 am. Differences in *Anopheles* mosquito densities among the trapping methods were compared using a generalized linear model based on a negative binomial distribution.

**Results**. There were 82.80% (2,136/2,580) of recaptures of wild-caught and 94.50% (2,835/3,000) of laboratory-reared mosquitoes that were molecularly identified as *An. harrisoni* and *An. minimus*, respectively. Mean density of *An. harrisoni* was significantly lower in HNDT (15.50 per night, 95% CI [12.48–18.52]) relative to HLC (25.32 per night (95% CI [22.28–28.36]), $p < 0.001$). Similarly, the mean density of a laboratory strain of *An. minimus* recaptured in HDNT was significantly lower (37.87 per night, 95% CI [34.62–41.11]) relative to HLC (56.40 per night, 95% CI [55.37–57.43]), $p < 0.001$. Relative sampling efficiency analysis showed that HLC was the more efficient trap in collecting the *An. minimus* complex in the SFS.

**Conclusion**. HDNT caught proportionately fewer *An. minimus* complex than HLC. HDNT was not sensitive nor significantly correlated with HLC, suggesting that it is not an alternative method to HLC.

## BACKGROUND

Globally in 2019, there were an estimated 227 million malaria cases reported in 85 malaria-endemic countries. In 2020, during the COVID-19 pandemic, the estimated number of malaria cases rose to 241 million cases across 108 countries that were malaria-endemic (*WHO, 2021*). In the Southeast Asia region, nine malaria-endemic countries, comprising Timor-Leste, Myanmar, Thailand, Bhutan, Bangladesh, Nepal, Democratic People's Republic of Korea, Indonesia, and India contributed about 5% of the overall malaria burden in 2020, representing a reduction of 78% from about 23 million in 2000 to five million in 2020 (*WHO, 2021*). Malaria continues to be a significant cause of morbidity and mortality in some malaria foci, especially along international border areas (*WHO, 2020*). As treatment failure rates for frontline anti-malarial drugs continue to worsen (*Amaratunga et al., 2016*; *Ashley et al., 2014*), control efforts focusing on malaria vectors have become increasingly important.

Among the 41 dominant vector species (*Sinka et al., 2011*), *Anopheles minimus* s.l. has been regarded as one of the efficient primary malaria vectors exhibiting heterogeneity of behavior in Southeast Asia (*Tananchai et al., 2019a*; *Tananchai et al., 2019b*; *Trung et al., 2005*; *Trung et al., 2004*). Different populations of *An. minimus* observed in various localities also differ in their endophilic and endophagic tendencies (*Trung et al., 2005*). The *An. minimus* complex (Theobald, 1901) comprises at least three formerly named sibling species, *An. minimus* former (species A), *An. harrisoni* former (species C), and *An. yaeyamaensis* former (species E) of which two species, *An. minimus* and *An. harrisoni* are sympatric in three villages in Kanchanaburi and Chiang Mai provinces, Thailand (*Tainchum et al., 2015*; *Tananchai et al., 2019a*). Sibling species of the Minimus complex comprise outdoor host-seeking *An. minimus* and *An. harrisoni* that exhibit zoophilic and variable endophilic (*Ismail, Phinichpongse & Boonrasri, 1978*; *Rwegoshora et al., 2002*), exophilic and exophagic behaviours (*Sungvornyothin et al., 2006*; *Tananchai et al., 2019b*) and opportunistic host seeking preferences (*Sinka et al., 2011*; *Sungvornyothin et al., 2006*). The frequency of indoor biting (*Ismail, Phinichpongse & Boonrasri, 1978*) and human biting (*Suthas et al., 1986*) both decreased following DDT spraying which may be explained either by the differential biting behaviors of Minimus complex species (*Garros et al., 2005*; *Tananchai et al., 2019a*) or shifts in species composition (*Carnevale & Manguin, 2021*; *Durnez & Coosemans, 2013*; *Garros et al., 2005*) and feeding and resting behavior over time (*Durnez & Coosemans, 2013*; *Rwegoshora et al., 2002*). Behavioural heterogeneity of anthropophily and zoophily with varying biting periods was reported in the Greater Mekong subregion (GMS) (*Kwansomboon et al., 2017*; *Manh et al., 2010*; *Trung et al., 2005*).

Differences in responses to insecticides can result in diverse exposure rates of species or subpopulations of the Minimus complex to the insecticide. For example, *An. minimus* showed very strong repellency responses to several insecticides and would have a higher survival chance in the presence of insecticides compared to *An. harrisoni*, which shows a much lower repellency response (*Potikasikorn et al., 2005*). Also, indoor residual spray (IRS) caused a shift to earlier and outdoor biting in Thailand (*Ismail, Phinichpongse & Boonrasri, 1978*), whereas on the other hand, in the foothills where *An. minimus* s.l. was

the main vector, no effect of DDT was seen on the already early-biting *An. minimus* s.l. population (*Ismail, Phinichpongse & Boonrasri, 1978*). Widespread use of IRS resulted in different behavior by *An. minimus* s.l. for example, marked zoophily compared to villages with lower DDT pressure in Thailand (*Nustsathapana et al., 1986*) which probably reflects a species shift from *An. minimus* to *An. harrisoni*, as also observed in Vietnam due to the widespread use of insecticide treated nets (*Garros et al., 2005*).

Monitoring these diverse behaviors requires special attention as little is known about the responses of these vectors to novel control measures, such as volatile pyrethroids with airborne effects. Next-generation vector surveillance tools are needed to monitor these behaviors for more cost-effective and successful malaria control (*Farlow, Russell & Burkot, 2020*; *WHO, 2019*). The traditional human landing catch method (HLC) is still regarded as the standard reference method for sampling host-seeking malarial mosquitoes (*Burkot et al., 2019*; *Lima et al., 2014*), requiring all-night supervision of trained collectors who are exposed to potentially infective mosquitoes (*Gimnig et al., 2013*; *Kilama, 2010*). A safer collection option is the human double net trap (HDNT) method, consisting of two untreated white box-type nets, with the inner net touching the ground protecting the human 'bait' and a second larger net which is placed directly over the inner net. The outer net is raised off the ground so that mosquitoes attracted to the human-bait are collected between the two nets. When the same person is used both as bait and collector, the collection effort is reduced, as well as the exposure to infectious mosquito bites during a short 15 min collection in the space between the inner and outer nets. Recent field evaluations of HDNT tested against the HLC method in Lao PDR and Ethiopia showed that the HDNT collected similar numbers of *Anopheles* as the HLC (*Degefa et al., 2020*; *Tangena et al., 2015*). In the Lao evaluation, the HDNT method collected a greater diversity of mosquito species than HLC, and both the HLC and HDNT capture rates were comparable at both high and low mosquito densities (*Tangena et al., 2015*). Validating alternative trap types to measure the human landing rate in the field and in SFS are needed prior to examining the effectiveness of alternative trap types for assessing the impact of personal protection interventions on mosquito landing (Neil Lobo, pers. comm.). However, whether the HDNT method could replace HLC for measuring human landing rates in an SFS has not yet been explored in the Asia-Pacific region. As rigorous data of performance of the NextGen vector surveillance tools are required for the target product profile, it is desirable to evaluate and describe the performance characteristics of the HDNT method in various eco-epidemiological settings following the framework of *Farlow, Russell & Burkot (2020)*. Thus, the aim of the current study was to compare the efficacy of an alternative surveillance method (HDNT) relative to HLC against laboratory and wild-caught strains of *An. minimus* in an SFS system.

## MATERIALS AND METHODS

### *Anopheles minimus* : laboratory (L) strain

*Anopheles minimus* s.s. (KU) originated in Rong Klang district, Prae province, northern Thailand in 1993, and was maintained at a field insectary for the semi-field experiment in Pu Tuey village (14°20′N; 98°59′E) following standard handling procedures and conditions

(25 ± 2 °C, 80 ± 10% relative humidity and 12:12 h daylight:darkness cycle) (*Boonyuan et al., 2017*; *Chareonviriyaphap, Prabaripai & Bangs, 2004*). Larval food (TetraMin®, TetraGmbH, Germany) was provided three times daily. Pupae were harvested daily, placed in small holding cups and the adults were allowed to emerge in wire-mesh cages (30 × 30 × 30 cm) where they were provided *ad libitum* with 10% sucrose (w/v) solution. Female adult mosquitoes were denied sucrose solution and provided with a water-soaked cotton pad ∼12 h before blood feeding. An artificial membrane feeding technique using human whole blood was used to maintain the self-mating mosquito colony (*Phasomkusolsil et al., 2013*). The pathogen-free blood supplied by Thai Red Cross Society was handled in the KU insectary following a written standard operating procedure (*Sukkanon et al., 2020*).

### *Anopheles minimus* s.l. wild (W) strain

Feral populations of *An. minimus* s.l. were collected using the double-cow net trap design. Briefly, a cow was tethered inside the inner of a double-net (inner: 2.5 m H × 3 m L × 3 m W, outer: 2 m H × 5 m L × 5 m W) (*Laurent et al., 2016*; *Sukkanon et al., 2021*)—similar to the human-baited double net design of *Tangena et al. (2015)*—and served as a bait for unfed mosquito collections. Mosquitoes resting on the interior walls of the double-cow net trap were collected using mouth aspirators by a well-trained local collector from 18:00 to 24:00 h for 15 min each hour. Collected mosquitoes were held in plastic holding cups topped with a cotton pad soaked with 10% sucrose solution and transferred to a field insectary located 50 m from the collection site for morphological identification (*Rattanarithikul et al., 2006*). Identified mosquitoes were deprived of sucrose and provided with a water-soaked cotton pad for approximately 6 h prior to testing. One hundred active *An. minimus* s.l. mosquitoes were released per replicate (night) and 25 unfed mosquitoes were held in the field insectary as the control for 12 h.

### Semi-field screen system

A semi-field screen house system (SFS) measuring 40 m L × 3.5 m H × 4 m W was supported by metal frames on a concrete block foundation with corrugated iron roofing (Fig. 1A). Several entry points and internal sliding double doors facilitated movement between the chambers by the data collectors. The enclosure could be modified into as many as four separate chambers using collapsible screen partitions (*Sukkanon et al., 2021*). In this study, two compartments were used with designated as chambers 'a and d', 30 m apart for the placement of HDNT and HLC. The floor in both rooms was lined with white plastic sheeting to facilitate observations and the recovery of knockdown mosquitoes (*Salazar et al., 2012*); each room was covered with an untreated mosquito net (2 m H × 10 m H × 4 m W, mesh size 1.5 mm) to prevent mosquitoes from escaping.

### Mosquito trap collections

The experiment was conducted during the hot season from early March to June, 2021. On the first day of the trial, the HDNT and HLC traps were randomly assigned to either end of SFS rooms 'a and d', each containing an untreated mosquito net. Collectors wore shorts up to the knee and a long-sleeve shirt and refrained from smoking, alcohol consumption and washing with soap (Fig. 1B). All collectors were trained and signed the consent form

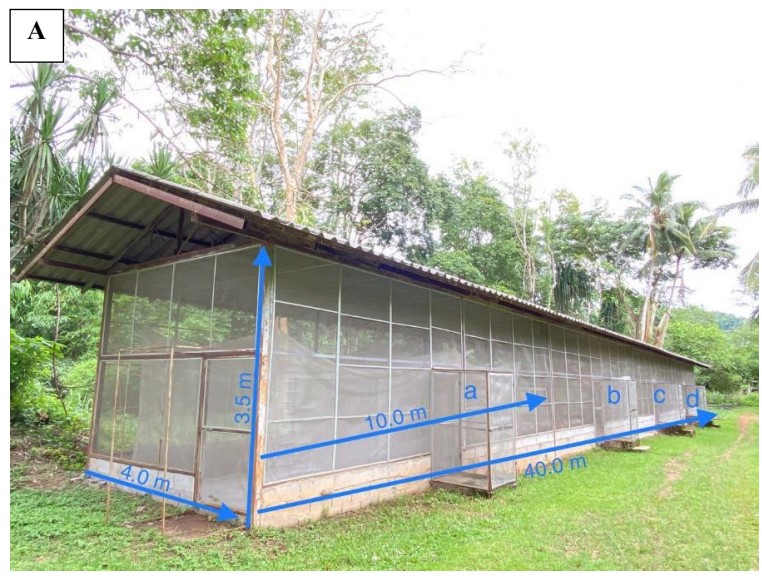

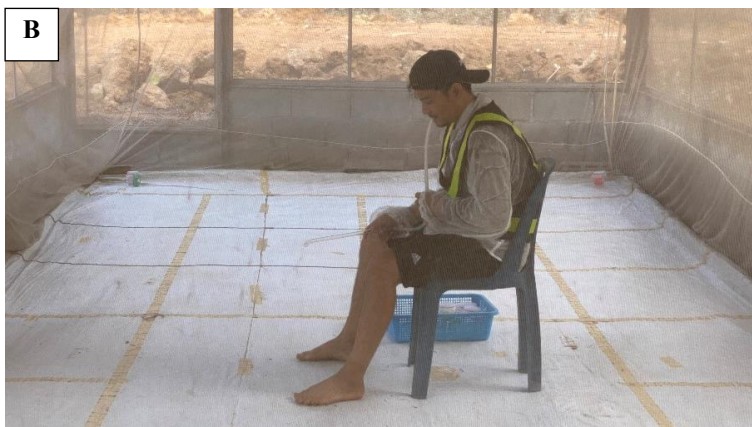

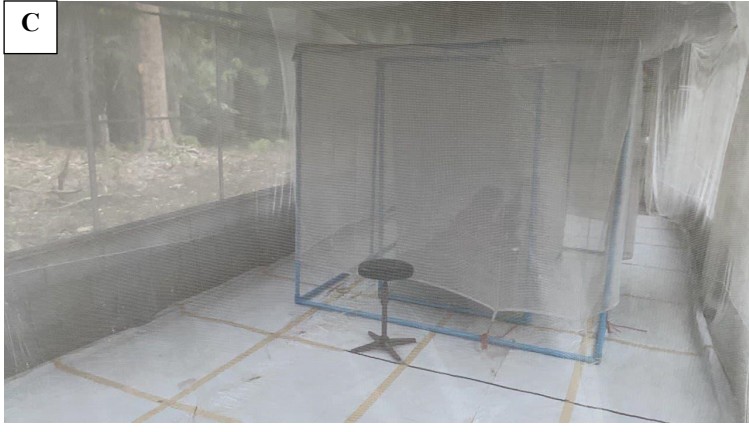

**Figure 1** **Experimental facility and traps.** (A) Semi-field system consists of four compartments (a, b, c, and d). (B) A collector performing human landing catches (HLC). (C) Human double net trap (HDNT) setup using PVC pipe.

prior to the study. Mosquitoes attracted to the human bait and resting on the interior walls of the two nets were collected by trained collectors (who acted as bait and the collector).

## Experimental design

Human landing catch (HLC) was performed by a healthy adult volunteer, acting as both bait and the collector (*Chareonviriyaphap et al., 2003*). Briefly, collectors sat on a chair with the lower limbs exposed from foot to knee and collected released mosquitoes landing on the legs of the collector before the mosquitoes commenced feeding. The human double net trap (HDNT) consisted of two stitched inner and outer nets. The inner net (97 cm H × 200 cm L × 100 cm W) protected a human volunteer who served as the bait and slept on a mattress (Fig. 1C). The outer net (100 cm H × 250 cm L × 150 cm W, mesh size 1.5 mm) was stitched to the inner net and raised 30 cm off the ground. For ease of installation, blue PVC pipe (outer diameter 25 mm) was used as the frame and pole to hang the double net.

After mosquitoes had been released in both rooms (50 mosquitoes per room) at 17:30 h, two trained local collectors (one each in the middle of the net in Rooms A and D) sat performing HDNT for 10 mins every hour and HLC for 50 mins every hour, respectively. Volunteers performed HLC or HDNT collections uninterrupted hourly using a flashlight and mouth aspirator to collect the specimens (*WHO, 2013*) from 18:00 to 06:00 h, 12 h per night with regular supervision. Ambient temperature (°C) and relative humidity (% RH) were recorded daily using a HOBO data logger at hourly intervals from 18:00 to 06:00 h. Recaptured landing mosquitoes were placed in separate labelled holding cups specific for HDNT and HLC in rooms A or D for each collection period, provided with 10% sucrose solution and held at optimum temperature and humidity conditions for 24 h of mortality observation in the field insectary. During the breaks (10 min), volunteers collected any knockdown/dead mosquitoes on the plastic sheet floor. At 06:00 h, the volunteers collected the remaining mosquitoes resting in both rooms with Prokopack aspirators and kept them in separate, clean, labelled cups. The HDNT and HLC collection methods were rotated between Rooms A and D every alternate night and volunteers were also rotated between the two traps nightly. Thirty consecutive replicates were conducted for releases of wild-caught *An. minimus* s.l. or lab-reared *An. minimus*.

## Data analysis

To compare the efficacy of HDNT with HLC, the number of recaptured landing mosquitoes by HLC was adjusted by multiplying by 1.2 (60/50 min). However, numbers of recaptured resting mosquitoes in HDNT were not adjusted as the resting collection was independent of catching effort.

The mean densities of mosquitoes (landing and resting) caught per night using HDNT were compared with those caught from HLC using the summation of the raw number of recaptured mosquitoes per trap at quarterly intervals (18:00 –21:00, 21:00–0:00, 0:00–03:00, and 03:00–06:00 h, per night for analysis using the Mann Whitney $U$ test (Wilcoxon Rank Sum test). The Kruskal–Wallis $H$ test for multiple comparisons was used to compare mean mosquito densities recaptured every quarterly. Data were summarized and reported as numbers of mean recaptured mosquitoes with 95% confidence intervals. The box-and-whisker plot in the SPSS software package compared the median, first and third quartiles,

minimum and maximum of mosquitoes recaptured quarterly per night. The abundance of mosquitoes landing in HDNT and HCL were pooled and analyzed for correlation with mean temperature and relative humidity using Spearman's correlation coefficient.

To determine whether HDNT correlated with the reference method (HLC), the log10 $(x + 1)$ transformations of the total numbers of mosquitoes caught by the alternative methods were analyzed using Spearman's correlation coefficient. As the number released per night was 100 and not considered as a 'sparse' mosquito count, the relative sampling efficiency (RSE) was estimated from the log ratio of the total number of mosquitoes caught by HDNT to the number caught by the reference method HLC ($\log_{10}(\text{HDNT} + 1) - \log_{10}(\text{HLC} + 1)$) plotted against the average mosquito abundance, calculated as $[\log_{10}(\text{HDNT} + 1) + \log_{10}(\text{HLC} + 1)]/2$ (*Hollis, 1996*; *Smith, 1993*). Simple linear regression analysis was performed to determine the relationship between the RSE and the average mosquito abundance. The coefficient of determination ($R^2$) derived from the analysis was interpreted as an estimate of the proportion of deviation from perfect linear correlation due to density-dependence rather than random error, with a high and significant value indicating density-dependence (*Degefa et al., 2020*). To test whether the RSE depended on mosquitoes, the mean log ratio and its antilog geometric mean ratio were calculated to estimate conversion factors between the HDNT and the HLC reference method for mosquito species (*Kenea et al., 2017*).

The influences of trapping method, volunteer and mosquito strains, as well as their interactions, on mosquito density, were also analyzed using the log likelihood ratio test (LRT). The difference in *Anopheles* mosquito density between the different trapping methods was compared using a generalized linear model (GLM) based on a negative binomial distribution. The number of mosquitoes was fitted as the independent variable, whereas, trap type, volunteers, chambers and mosquito strain were treated as the independent variables in the evaluation. The estimated marginal mean (EMM) density of *Anopheles* mosquitoes was determined for each trap using the negative binomial regression with adjustment for volunteer and species strain. Bonferroni corrections for multiple comparisons were performed to determine the statistical significance of differences in numbers of nightly mosquito recaptures among traps. The actual number of recaptured mosquitoes (unadjusted number of mosquitoes in the HLC) were used in this analysis. Data were analyzed using the SPSS version 20.0 (SPSS, Inc., Chicago, IL, USA) statistical package. All levels of statistical significance were set at 5% ($p < 0.05$).

## Molecular identification of *An. minimus* sensu lato

DNA extraction was performed individually on a 10% (251/2,580) sample of the morphologically identified *An. minimus* complex from the outdoor collection of double-cow net trap based on the protocol of (*Manguin et al., 2002*). Multiplex allele-specific polymerase chain reaction assay was performed for molecular species identification within the Minimus complex and related species following the protocol of (*Sungvornyothin et al., 2006*). Briefly, in a final volume of 25 µl, the polymerase chain reaction (PCR) amplification conditions were: 1X reaction buffer, 0.2 mM of dNTPs, 1.5 mM of MgCl2, 0.2 µM of each primer, 2.5 units of Taq DNA polymerase (Invitrogen), and 1 µl of DNA template. The

PCR cycles were: one cycle at 94 °C for 2 min, follow by 40 cycles of a denaturation step at 94 °C for 30 s, annealing at 50 °C for 30 s, and extension at 72 °C for 40 s, and a final extension at 72 °C for 5 min. Lastly, the PCR product was subjected to electrophoresis on a 3% agarose gel at 100 V for 30 min and stained with ethidium bromide.

### Ethics approval and consent to participate

Ethics approval for the study was provided by the Research Ethics Review Committee for Research Involving Human Research Participants, Kasetsart University (Certificate of Approval No. CAO63/035). Formal ethical clearance of the study protocol and volunteer collector informed consent were obtained before commencing the trials. For the HLC method, assisted guidance was provided to ensure probing mosquitoes were collected prior to biting, including information, awareness of the risks of wild mosquitoes in pathogen transmission, and a guarantee of medical care for the duration of the study.

## RESULTS

### Estimation of variables from general linear modeling

The mean abundance of *Anopheles* species varied significantly by trap type (LRT $X^2 = 39.46$, $p < 0.001$), volunteer (LRT $X^2 = 25.98$, $p < 0.001$), mosquito strains (LRT $X^2 = 403.23$, $p < 0.001$), and across trap types and volunteer (LRT $X^2 = 5.42$, $p = 0.020$), as shown in Table 1.

Overall, HLC yielded 27.00% (95% CI [1.21–1.43], $p < 0.001$) more anophelines compared to HDNT (Table 2, Fig. 2C). Significantly, 21.00% (95% CI [0.75–0.88], $p < 0.001$) more anophelines were attracted to volunteer 1 relative to volunteer 2 (Tables 2 and 3, Fig. 2B). A significantly higher abundance level of 85.00% (95% CI [2.15–2.55], $p < 0.001$) of the laboratory strain (*An. minimus*) was collected in both traps compared to wild strains (*An. harrisoni*), as shown in Table 2 and Table S1. However, chamber did not affect the experiment ($p = 0.202$), as shown in Table 2 and Fig. 2A.

### Semi-field trials with wild strain of *An. harrisoni*
#### Recapture rate

Overall, 2,580 female *An. minimus* s.l. mosquitoes comprising *An. harrisoni* (82.80%), *An. minimus* (15.30%), *An. aconitus* (0.015%), and *An. varuna* (0.004%) were collected for 30 nights using the HDNT and HLC methods. The total numbers of *An. harrisoni* recaptured in landing collections in HDNT and HLC were 465 (15.50%) and 777.60 (26.80%), respectively. The lower numbers of *An. harrisoni* ($n = 2136$) were significantly recaptured using HDNT (15.50 per night, 95% CI [12.48–18.52]) relative to HLC (25.32 per night (95% CI [22.28–28.36]), $p < 0.001$ (Table 4).

#### Resting rate

The total numbers of resting *An. harrisoni* recaptured using Prokopack aspirators in the HDNT and HLC rooms were 804 (26.80%) and 678 (22.60%), respectively (Table 4 and Table S1). The mean temperature and relative humidity during the trials were in the ranges 28.78–26.67 °C and 79.87–91.11%, respectively, from 18:00 to 00:00 h and 25.56–24.94 °C and 94.91–96.78%, respectively, from 00:00 to 06:00 h, (Table 5). Overall, remaining

**Table 1** Loglikelihood ratios for two *Anopheles* species whose abundance levels were influenced by trap, volunteer, and mosquito strains, and interaction between traps-volunteermosquito strains.

| Variable | $\chi^2$ | df | *p*-value |
| --- | --- | --- | --- |
| Trap (HLC - HDNT) | 39.46 | 1 | <0.001 |
| Chamber (a - d) | 1.63 | 1 | 0.202 |
| Volunteer (1–2) | 25.98 | 1 | <0.001 |
| Mosquito strain (W - L) | 403.23 | 1 | <0.001 |
| Trap * Chamber | 0.80 | 1 | 0.372 |
| Trap * Volunteer | 5.42 | 1 | 0.020 |
| Trap * Mosquito strain | 1.15 | 1 | 0.283 |

Notes.
HLC, human landing catch; HNDT, human double net trap; W, wild; L, laboratory.

**Table 2** Parameter estimates of mosquito collection from semi-field system (SFS).

| Variable | Effect | Estimate | SE | OR | 95% Exp(B) Confidence interval | | z-score | *p*-value |
| --- | --- | --- | --- | --- | --- | --- | --- | --- |
| | | | | | Lower | Upper | | |
| (Intercept) | (Intercept) | 3.32 | 0.02 | 27.43 | 26.28 | 28.61 | 152.27 | <0.001 |
| Trap | HLC - HDNT | 0.27 | 0.04 | 1.31 | 1.21 | 1.43 | 6.26 | <0.001 |
| Chamber | a - d | −0.05 | 0.04 | 0.95 | 0.87 | 1.03 | −1.28 | 0.202 |
| Volunteer | 2 - 1 | −0.21 | 0.04 | 0.81 | 0.75 | 0.88 | −5.09 | <0.001 |
| Mosquito | L - W | 0.85 | 0.04 | 2.34 | 2.12 | 2.55 | 19.61 | <0.001 |
| Trap * Chamber | HLC - HDNT * a - d | −0.07 | 0.08 | 0.93 | 0.79 | 1.09 | −0.89 | 0.372 |
| Trap * Volunteer | HLC - HDNT * 1 - 2 | 0.19 | 0.08 | 1.21 | 1.03 | 1.43 | 2.33 | 0.020 |
| Trap * Mosquito | HLC - HDNT * L - W | −0.09 | 0.08 | 0.91 | 0.77 | 1.08 | −1.07 | 0.284 |

Notes.
HLC, human landing catch; HNDT, human double net trap; W, wild; L, laboratory.

mosquitoes caught resting in both rooms had a significantly higher recapture rate of *An. harrisoni* in HDNT (26.80 per night, 95% CI [24.20–29.40]) than HLC (22.60 per night, 95% CI [20.06–25.13]), $p = 0.026$ (Table 4).

### Quarterly collections

Quarterly night collections showed significantly lower numbers of *An. harrisoni* in HDNT (1.73 per night, 95% CI [1.08–2.39]) relative to HLC (5.72 per night, 95% CI [4.13–7.31]), $p < 0.001$ during 18:00–21:00 h. A similar pattern was seen during 21:00–00:00 h for HDNT (4.37 per quarter, 95% CI [3.10–5.64]) relative to HLC (9.44 per quarter, 95% CI [7.61–11.27]), $p < 0.001$. No significant difference in the mean density of *An. harrisoni* was seen during 00:00–03:00 h between HDNT (5.20 per quarter, 95% CI [3.66–6.74]) and HLC (6.88 per quarter, 95% CI [5.53–8.23]), $p = 0.055$, and during 03:00–06:00 h between HDNT (4.20 per quarter, 95% CI [3.13–5.27]) and HLC (3.28 per quarter, 95% CI [2.10–4.46]), $p = 0.160$), as shown in Table 5 and Fig. 3A.

### Correlation between HDNT and HLC

Spearman's correlation coefficient for the relationship between HDNT and HLC of landing and resting *An. harrisoni* caught per trap showed weak, negative relationships with r =

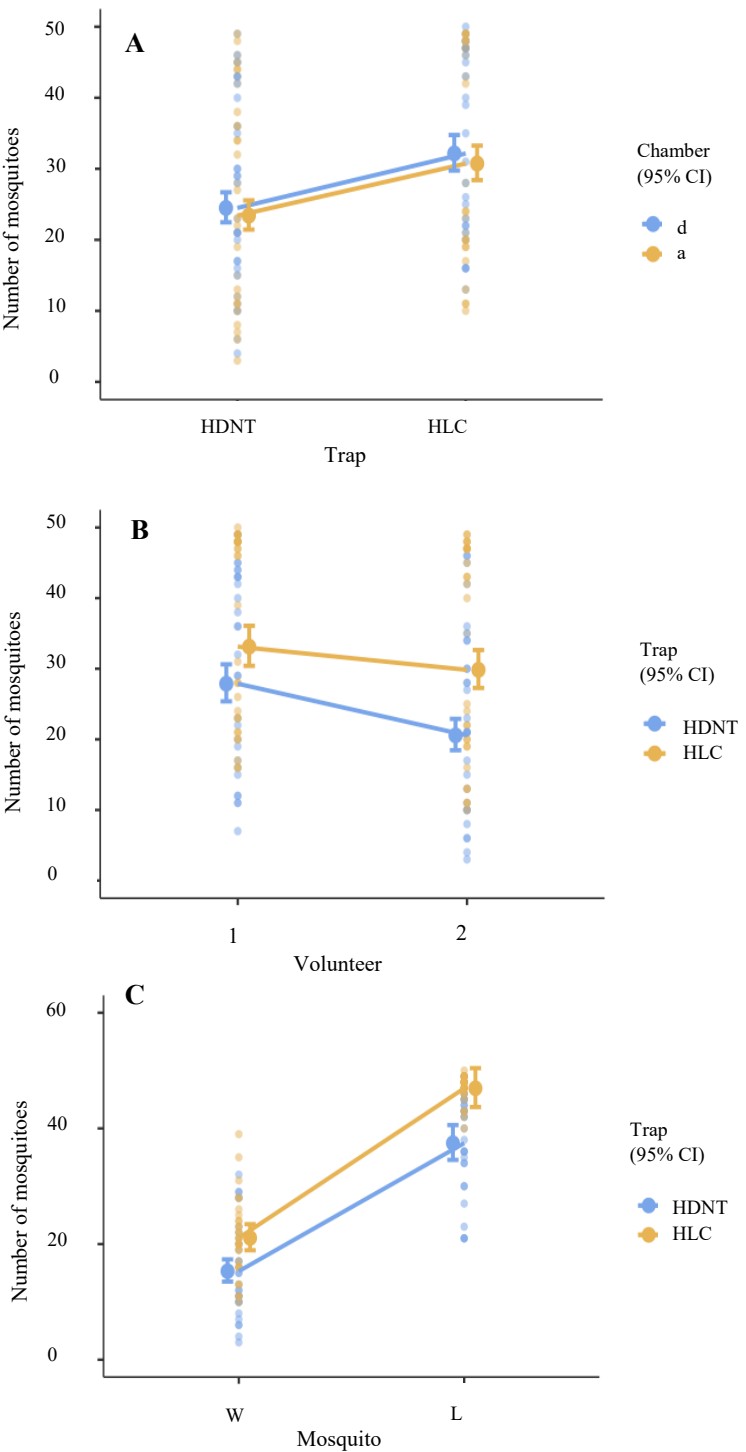

**Figure 2  Loglikelihood ratio tests.** (A) Total numbers of mosquitos collected using human double net trap (HDNT) and human landing catches (HLC) within chambers. (B) Number of mosquitoes recaptured by volunteers 1 and 2 within the traps. (**C**) Prediction of different mosquito strains on traps. W, wild strain; L, laboratory strain.

**Table 3** Estimates of a negative binomial regression for comparison of semi-field system (SFS) host-seeking anopheline density between human landing catches (HLC) and human double net trap (HDNT) and volunteers.

| Species (strain) | Traps | Number collected | EMM (95% CI) | OR | *p*-value |
|---|---|---|---|---|---|
| *An minimus* (L) | HDNT | 1,136 | 37.87 (34.62-41.11)[a] | 0.79 | <0.001 |
| | HLC | 1,410 | 47.09 (32.79-67.63)[b] | 1.00 | |
| *An harrisoni* (W) | HDNT | 465 | 15.50 (12.48-18.52)[a] | 0.73 | <0.001 |
| | HLC | 633 | 20.90 (14.48-30.15)[b] | 1.00 | |
| **Two species** | | | | | |
| Volunteer 1 | HDNT | 681 | 20.50 (18.70-22.50)[a] | 0.69 | <0.001 |
| | HLC | 970 | 29.60 (27.30-32.10)[b] | 1.00 | |
| Volunteer 2 | HDNT | 920 | 27.90 (25.70-30.30)[a] | 0.84 | 0.015 |
| | HLC | 1,073 | 33.30 (30.80-36.00)[b] | 1.00 | |

Notes.
HLC, human landing catch; HNDT, human double net trap; W, wild; L, laboratory.
Different lowercase superscripts (a and b) in columns indicate significant differences between groups using Bonferroni Post Hoc test ($p < 0.05$).

**Table 4** Mean of recaptured mosquitoes landing and resting in rooms set up for human double net trap (HDNT) and human landing catches (HLC) methods in semi-field system (SFS).

| Molecular species | Preferences of recapture | Night | HDNT room | | HLC room | | z-score | p-value[**] |
|---|---|---|---|---|---|---|---|---|
| | | | Total recaptured[*] | Mean recaptured/ night (95% CI) | Total recaptured[*] | Mean recaptured/ night (95% CI) | | |
| *An. harrisoni* | Landing | 30 | 465 | 15.50 (12.48–18.52) | 777.60 | 25.32 (22.28–28.36) | −4.02 | <0.001 |
| *An. minimus* | Landing | 30 | 1,136 | 37.87 (34.62–41.11) | 1,692 | 56.40 (55.37–57.43) | −6.64 | <0.001 |
| *An. harrisoni* | Resting | 30 | 804 | 26.80 (24.20–29.40) | 678 | 22.60 (20.06–25.13) | −2.23 | 0.026 |
| *An. minimus* | Resting | 30 | 263 | 8.77 (5.76–11.77) | 26 | 0.87 (0.35–1.38) | −5.11 | <0.001 |

Notes.
HLC, human landing catch; HNDT, human double net trap.
[*]Total number of mosquitoes recaptured by volunteers for 30 nights in each trap.
[**]Mann Whitney $U$ test ($p < 0.05$).

−0.02, $p = 0.926$ and r = −0.08, $p = 0.672$, respectively (Figs. 4A & 4E, Table S3). The $R^2$ values of HDNT *versus* HLC in landing recapture ($R^2 = 0.24$, $p = 0.006$) suggested that the RSEs of the HDNT was dependent on landing mosquito density, but resting recapture ($R^2 = 0.09$, $p = 0.109$) was not dependent on mosquito density (Figs. 4B & 4F, Table S3). The mean log ratios of HDNT compared to HLC in landings caught were negative, suggesting that the alternative trap was less effective than HLC in collecting *An. harrisoni*. However, the mean log ratio was positive in recapture of resting mosquitoes remaining in the SFS room. Based on the geometric mean ratios (GMR), the number of catches from HDNT was the same as that of HLC from both landing and resting collections (Table S4).

### Association with ambient temperature and relative humidity
Data from both trapping methods were pooled to determine the interaction between temperature, relative humidity and quarterly mosquito density. *An. harrisoni* abundance

Peer J

**Table 5  Mean of quarterly recaptured landing mosquitoes in human double net trap (HDNT) and human landing catches (HLC) for 30 nights per quarter in SFS.**

| Molecular species | Time of night | Mean temperature, (°C) (95% CI) | Mean relative humidity (%), (95% CI) | HDNT | | HLC | | z-score | P-value[***] |
|---|---|---|---|---|---|---|---|---|---|
| | | | | Total recaptured[*] | Mean recaptured/ quarter h[**] (95% CI) | Total recaptured | Mean recaptured/ quarter h[**] (95% CI) | | |
| *An. harrisoni* | Quarter I (18:00–21:00) | 28.78 (28.26–29.31) | 79.87 (74.26–85.49) | 52 | 1.73 (1.08–2.39)[a] | 171.60 | 5.72 (4.13-7.31)[a,b] | −4.37 | <0.001 |
| | Quarter II (21:00–00:00) | 26.67 (26.18–27.15) | 91.11 (87.83–94.39) | 131 | 4.37 (3.10–5.64)[b] | 283 | 9.44 (7.61–11.27)[c] | −3.93 | <0.001 |
| | Quarter III (00:00–03:00) | 25.56 (25.08–26.05) | 94.91 (92.85–96.97) | 156 | 5.20 (3.66–6.74)[b] | 172 | 6.88 (5.53–8.23)[b,c] | −1.92 | 0.055 |
| | Quarter IV (03:00–06:00) | 24.94 (24.39–25.50) | 96.78 (95.33–98.23) | 126 | 4.20 (3.13–5.27)[b] | 98.40 | 3.28 (2.10–4.46)[a] | −1.40 | 0.160 |
| *An. minimus* | Quarter I (18:00–21:00) | 29.62 (29.25–29.98) | 84.82 (81.66–87.89) | 583 | 19.43 (16.97–21.90)[a] | 1369.20 | 45.64 (41.99–49.29)[a] | −6.49 | <0.001 |
| | Quarter II (21:00–00:00) | 28.39 (28.13–28.65) | 93.59 (91.50–95.27) | 314 | 10.47 (8.43–12.50)[b] | 267.60 | 8.92 (6.21–11.63)[b] | −1.10 | 0.270 |
| | Quarter III (00:00–03:00) | 27.90 (27.65–28.16) | 96.45 (95.19–97.72) | 172 | 5.73 (4.05–7.41)[c] | 50.40 | 1.68 (0.35–3.01)[c] | −4.63 | <0.001 |
| | Quarter IV (03:00–06:00) | 27.41 (27.16–27.65) | 98.00 (97.15–98.45) | 67 | 2.23 (1.30–3.17)[d] | 4.80 | 0.16(−0.67–0.39)[c] | −4.48 | <0.001 |

**Notes.**

HLC, human landing catch; HNDT, human double net trap; W, wild; L, laboratory.

[*]Total number of mosquitoes recaptured by volunteers over 30 nights in each trap.

[**]Different lowercase superscript letters (a, b, c, and d) in columns indicate significant differences between species-specific doses using Kruskal-Wallis H test for multiple comparisons ($p < 0.05$).

[***]Comparison mean of mosquito within row using Mann Whitney $U$ test ($p < 0.05$).

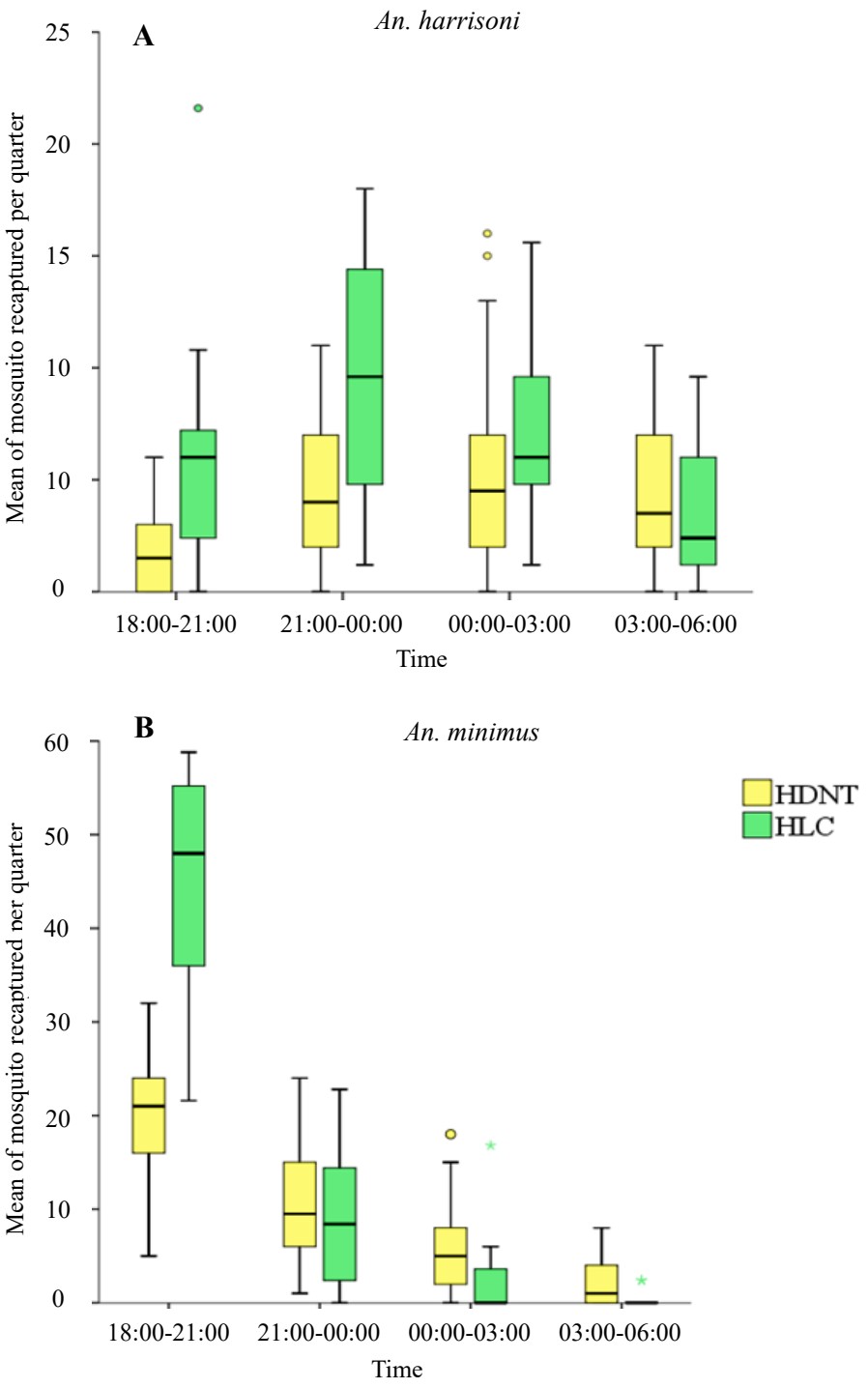

**Figure 3** **Mean of mosquitoes recaptured every quarter by both trap methods human double net trap (HDNT) and human landing catches (HLC).** (A) Wild strain *An. harrisoni*. (B) Laboratory strain *An. minimus*. The middle horizontal line of the box represents the median and the error bars represents the 95% confidence interval value of mosquitoes caught per trap per quarter. The circle and asterisk represent the outlier value of mosquitoes caught per trap per quarter.

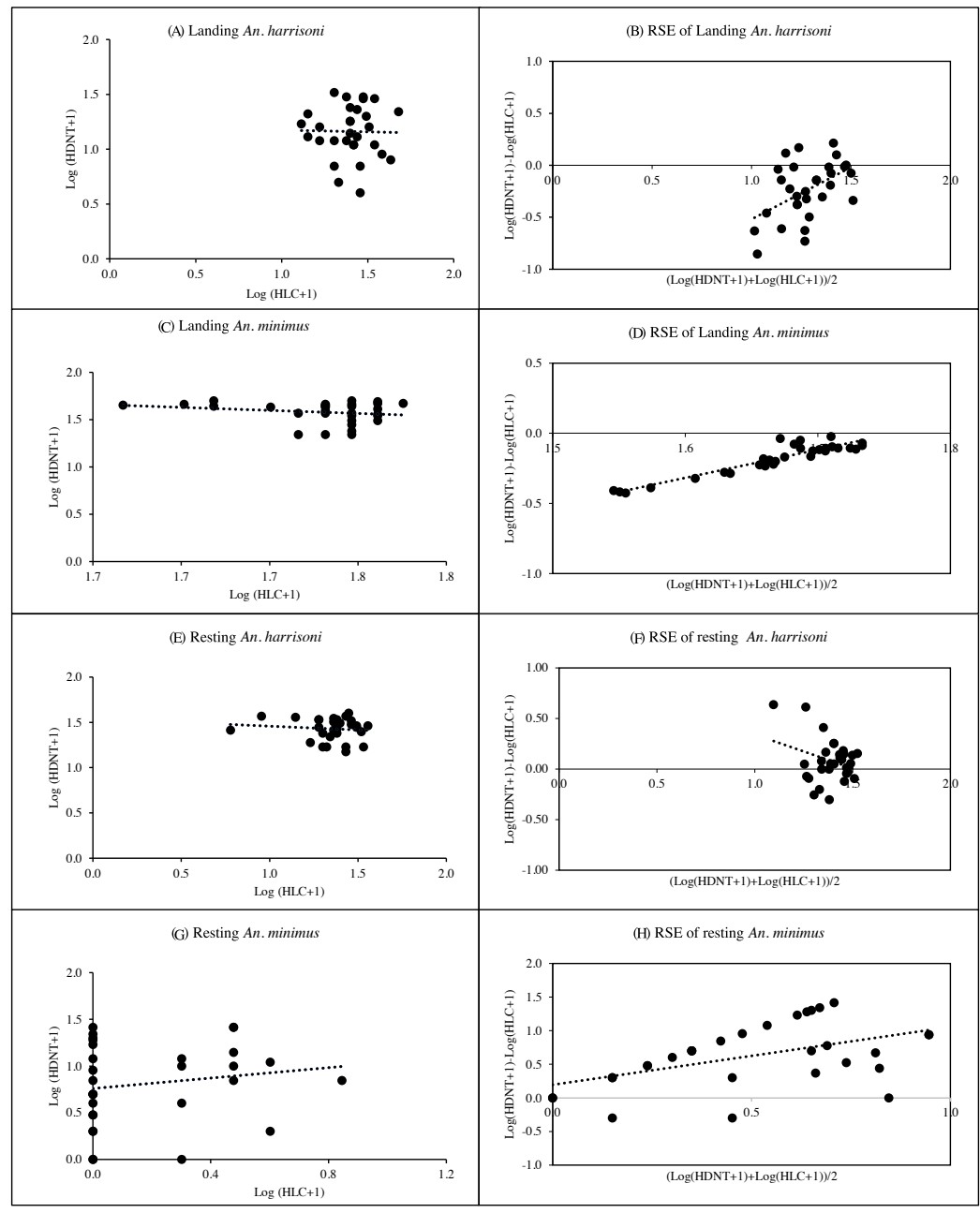

**Figure 4 Spearman's correlation coefficient for the relationship among log-transformed of landing and resting mosquitoes caught per trap.** (A) Landing *An. harrisoni*; (B) RSE of Landing *An. harrisoni*; (C) landing *An. minimus*; (D) RSE of landing *An. minimus*; (E) resting *An. harrisoni*; (F) RSE of resing *An. harrisoni*; (G) resting *An. minimus*; (H) RSE of resting *An. minimus*. (A, C, E, G) Correlation and density-dependence of human double net trap (HDNT) relative to human landing catches (HLC) for recapture of mosquitoes. (B, D, F, H) Relative sampling efficiency (RSE) of HDNT relative to HLC in collection of landing and resting *An. harrisoni* and *An. minimus*.

was not correlated with temperature (r = −0.08, p = 0.412), but had a weak, positive correlation with relative humidity (r = 0.23, p = 0.011), as shown in Table S5. Nightly abundance was correlated to increased relative humidity (r = 499, p = 0.005), but not affected by temperature (r = 0.09, p = 0.065), as shown in Table S6 and Figs. S1B, 1D.

### Semi-field trials with laboratory strain of *An. minimus*
#### *Recapture rate*
A high recapture rate of 94.50% of the laboratory strain *An. minimus* (n = 3,000 released) was recorded in HDNT and HLC over 30 nights (Table S2). The total numbers of landing *Anopheles* collected using HLC and in HDNT were 1,410 (47.00%) and 1,136 (37.87%), respectively (Table 3). The mean landing mosquito abundance recaptured in HDNT was significantly lower (37.87 per night, 95% CI [34.62–41.11]) relative to HLC (56.40 per night, 95% CI [55.37–57.43]), p < 0.001 (Table 4).

#### *Resting rate*
The total numbers of resting *An. minimus* recaptured using the Prokopack aspirators in the HLC and HDNT rooms were 26 (0.87%) and 263 (8.77%), respectively (Table 4). The mean temperature and relative humidity levels during the trials were in the ranges 29.62–28.39 °C and 84.82–93.59%, respectively, during 18:00–00:00 h and 27.90–27.41 °C and 96.45–98%, respectively, during 00:00–06:00 h (Table 5). Similarly, the mean abundance of resting *An. minimus* was significantly higher in HDNT (8.77 per night, 95% CI [5.76–11.77]) relative to HLC (0.87 per night, 95% CI [0.35–1.38]), p < 0.001 (Table 4).

#### *Quarterly collections*
Quarterly night collections show significantly lower numbers of resting *An. minimus* in HDNT during 18:00–21:00 h (19.43 per quarter, 95% CI [16.97–21.90]) relative to HLC (45.64 per quarter, 95% CI [41.99–49.29]), p < 0.001. No significant difference in the mean density of *An. minimus* was seen during 21:00–00:00 h between HDNT (10.47 per quarter, 95% CI [8.43–12.50]) and HLC (8.92 per quarter, 95% CI [6.21–11.63], p = 0.270). However, during 00:00–03:00 h, significantly higher recaptures were recorded of *An. minimus* in HDNT (5.73 per quarter, 95% CI [4.05–7.41]) relative to HLC (1.68 per quarter, 95% CI [0.35–3.01]), p < 0.001. A similar pattern was observed during 03:00–06:00 h in HDNT (2.23 per quarter, 95% CI [1.30–3.17]) and HLC (0.16 per quarter 95% CI [−0.67 to −0.39]), p < 0.001 (Table 5, Fig. 3B).

#### *Correlation between HDNT and HLC*
There was a weak, negative correlation between HDNT *versus* HLC for collection of landing *An. minimus* (r = −0.09, p = 0.636) and a weak positive correlation for resting collection (r = 0.17, p = 0.372) (Figs. 4C & 4G, Table S3). The R² values of HDNT *versus* HLC in landing recapture (R² = 0.87, p < 0.001) suggested that the RSEs of the HDNT were dependent on landing mosquito density, but resting recapture (R² = 0.03, p = < 0.329) was not dependent on mosquito density (Figs. 4D & 4H, Table S3). The mean log ratios of HDNT *versus* HLC in the landing catch was negative, suggesting that the alternative trap was less effective than HLC in collecting *An. minimus*. However, the mean log ratio was

positive in recapture of resting mosquitoes remaining in the SFS room. The mean GMR for catches from HDNT was 0.063 and the same as that from HLC in landing and resting collections, respectively (Table S4).

***Association with ambient temperature and relative humidity***

Data from both trapping methods were pooled to determine the interaction between mosquito density, temperature, and relative humidity. Abundance of *An. minimus* was significantly positively correlated with nightly temperature ($r = 0.61$, $p < 0.001$); however it was significantly negatively correlated with relative humidity ($r = -0.58$, $p < 0.001$), as shown in Table S5. No association of nightly abundance of *An. minimus* was observed between a decrease in temperature ($r = 0.13$, $p = 0.482$) and relative humidity ($r = 0.16$, $p = 0.394$), respectively (Table S6, Figs. S2B, S2D).

## DISCUSSION

This study evaluated the feasibility of estimating conversion factors between an exposure-free alternative to the HLC method for the surveillance of outdoor, host-seeking, Asian malaria vectors. The ultimate aim was to examine if HDNT and HLC conducted in an SFS could provide reliable estimates of biting rates of humans by mosquitoes for entomological surveillance in the National Malaria Control Programmes as this study respond to a "wider array of surveillance methods" that is recommended for different vector behaviors, particularly outdoor host-seeking ones (*van de Straat et al., 2021*). The results, based on analyses by both simple regression analysis and GLM statistical approaches (*Altman & Bland, 1983*; *Hollis, 1996*; *Kenea et al., 2017*), indicated that reliable conversion factors between HDNT and HLC could not be calculated despite adequate mosquito densities in the SFS. The poor correlation between the two collection methods was not surprising. The GLM analysis confirmed that the percentage of nightly mosquito catches per trap was significantly affected by the HDNT method relative to HLC, as shown by the low values of the RSEs, GMRs and mean log ratios. The RSE was dependent on mosquito density since a fixed number of mosquitoes were released per night, resulting in a high recapture rate and a narrow, expected range of 95% individual ratios of HDNT:HLC. This result differed from another HLC:CDC light trap study in northern Thailand (*Somboon, 1993*) as the variations and the biases of mosquito collection could be controlled in the current study, such as an enclosed environment sharing similar climatic conditions to the natural ecosystem and the consistent use of standardized baited trapping methods with good supervision. In Thailand, sympatric populations of *An. minimus* and *An. harrisoni* are considered zoophilic with variable indoor and outdoor biting behaviors (*Durnez & Coosemans, 2013*; *Sungvornyothin et al., 2006*; *Tananchai et al., 2019b*) which suggests highly opportunistic habits and considerable plasticity in host selection (*Edwards et al., 2019*; *Sinka et al., 2011*) after the discontinuation of IRS since 2006 (*Garros et al., 2006*; *Sungvornyothin et al., 2006*). In the absence of an animal or alternative host, it is inevitable these mosquitoes would display a high degree of anthropophily in the SFS.

In the absence of a net in the SFS, *An. minimus* and *An. harrisoni* could easily seek humans and directly land on hosts performing HLC compared to the HDNT method,

as the latter has two nets separating the collector from the mosquitoes. However, with a human bait inside the HDNT, a large proportion of mosquitoes seeking a blood meal are deterred from entry due to the physical barrier, thus reducing the overall sensitivity of the HDNT because these nets interfere with host seeking behavior and underestimate the abundance of *An. minimus* and *An. harrisoni*. In Lao PDR, anthropophilic mosquitoes attracted to HDNT were seeking shelter and entering the bed net accidentally with greater numbers collected in HDNT compared to HLC (*Tangena et al., 2015*). However, HLC was more productive than other alternative traps (CDC-light traps, animal-baited traps, and host decoy traps) in south-central Ethiopia (*Kenea et al., 2017*), Indonesia (*St Laurent et al., 2018*) and South Sulawesi (*Davidson et al., 2020*) for sampling *Anopheles* species.

For more than four decades, HLC was the most frequently used and preferred sampling method (*Silver, 2008*; *van de Straat et al., 2021*) because it directly estimates the epidemiologically relevant indicators of exposure of humans to biting mosquitoes (*Farlow, Russell & Burkot, 2020*). The HLC method has other drawbacks: (1) extremely labor-intensive and the risk of exposing collectors to malaria (*Gimnig et al., 2013*) and arboviral pathogens; (2) high level of supervision to maintain quality and collector efficiency (*Lima et al., 2014*), and (3) natural human variations in attractiveness to mosquitoes, thus impacting the accuracy and representativeness of human exposure (*Briet et al., 2015*; *Lima et al., 2014*; *Tangena et al., 2015*). To a large extent some of these biases can be addressed by standardizing the mosquito age and sequential releases of single species of *An. minimus* complex in sufficient numbers in the SFS. Low *Anopheles* abundance in outdoor settings have undermined the evaluation of HDNT and HLC in Lao PDR (*Tangena et al., 2015*) and Vietnam (Ratchadawan, submitted); and CDC-light traps baited with $CO_2$ and octenol in Brazil (*Lima et al., 2014*).

HDNT was described by (*Gater, 1935*) and tested in Africa, Asia, and South America with varying outcomes (*Rubio-Palis & Curtis, 1992*; *Service, 1977*). In African settings dominated by Anophelines, HLC collected almost 2.9 times as many *An. gambiae* in Nigeria (*Service, 1963*) and 5.4 times as many *An. gambiae* in Cameroon (*Le Goff et al., 1997*) as did HDNT. In western Venezuela, HDNT trapped only three anophelines compared with 1,237 collected using HLC for 36 h in night surveillance. In fact, HDNT was discarded due to its poor efficiency in the collection of Anophelines compared with HLC (*Le Goff et al., 1997*; *Rubio-Palis & Curtis, 1992*). However, HDNT successfully captured 2.5 times more daytime host-seeking *Aedes albopictus* relative to HLC (HDNT 1093: HLC 428) in Shanghai, China (*Gao et al., 2018*). The number of *Anopheles* mosquitoes captured using HDNT was 1.2 times higher than for HLC in Lao PDR (*Tangena et al., 2015*). The current study produced similar findings as the African and Venezuela studies, with the number of recaptured *An. minimus* and *An. harrisoni* being significantly lower than for HLC by 1.5 and 1.7 times, respectively.

Given the lack of power to detect a relationship between total mosquito numbers caught using HDNT and HLC in Lao PDR (*Tangena et al., 2015*), and greater numbers of anthropophilic mosquitoes caught in HDNT compared to HLC, our study standardized the sample size by releasing a fixed number of mosquitoes ($n = 100$) per night in the SFS, thus enabling high recapture rates of 91.67% and 88.93% for HLC and HDNT,

respectively. Despite this, and the lack of a correlation between the two methods, we concluded that the variability of recaptured mosquitoes was attributable to several factors, such as mosquito species, seasonal variation, and weather conditions. For example, temperature and wind movement can influence mosquito activity patterns (*Bowen, 1991*), abundance (*Ram et al., 1998*; *Wu et al., 2007*), and survival (*Ciota et al., 2014*). An increase in environmental temperature was associated with increased host seeking activity of *Culex pipiens, Aedes detritus* and *Aedes caspius* (*Drakou et al., 2020*) and *Anopheles* populations (*Asgarian, Moosa-Kazemi & Sedaghat, 2021*). Therefore, it accelerated the blood digestion, increased human biting frequency, shortened the gonotrophic cycle of *An. minimus* and *An. balabacensis balabacensis* (now *An. dirus*) during the dry and cool seasons (*Ismail, Phinichpongse & Boonrasri, 1978*; *Ismail, Notananda & Schepens, 1975*) and the extrinsic incubation period  (*Shaw, Marcenac & Catteruccia, 2021*), and increased disease transmission efficiency (*Afrane et al., 2006*; *Githeko et al., 2000*) during the dry and cool season in a forested area of Saraburi province, Thailand (*Ismail, Phinichpongse & Boonrasri, 1978*).

The association between the quarterly and nightly pooled abundance of HLC and HDNT mosquitoes and climate variables varied due to the significant inverse correlation of *An. minimus* densities to the relative humidity only. There was a similar correlation reported between nightly abundance of *Anopheles* species with relative humidity (range: 27.00–56.40%) and ambient temperature (range: 29.20 °C–30.00 °C) during the peak biting period (*Asgarian, Moosa-Kazemi & Sedaghat, 2021*). Significant direct correlations of paired *An. harrisoni* density and relative humidity and *An. minimus* density and temperature associations were observed for quarterly pooled abundance in our study site and were consistent with a study in Saraburi province, Thailand (*Ismail, Phinichpongse & Boonrasri, 1978*) and of *An. gambiae s.l.* in southeastern Senegal (*Diallo et al., 2019*). A similar correlation of *An. harrisoni* density and relative humidity for nightly pooled abundance in Pu Tuey was observed in Bangladesh (*Bashar & Tuno, 2014*), Saudi Arabia (*Jemal & Al-Thukair, 2018*), and Sri Lanka (*Yasuoka & Levins, 2007*). Our study was consistent with other finding in Pu Tuey, western Thailand (*Chareonviriyaphap et al., 2003*; *Tisgratog et al., 2012*) and central Vietnam (*Edwards et al., 2019*), where a decrease in the *An. minimus* density from a peak in the first quarter (18:00–21:00) to the fourth quarter was associated with a mean temperature drop from 29.62 °C–27.41 °C, and with an increase in the mean relative humidity from 84.82% to 98%, respectively. However, there was no clear relationship between temperature (range: 22.00 °C–24.00 °C) and the number of *Anopheles* caught using outdoor HLC, which was due to heavy rainfall, as this factor is a predominant driver of low mosquito abundance during elevated relative humidity observed in October 2021 in Mondulkiri, Cambodia  (Neil Lobo, pers. comm., March 2022). Although a decrease in humidity corresponded with low median survival of *An. gambiae* by 5–7 days in western Kenya, mosquitoes exhibited an enhanced reproductive fitness by 40% over the course of their life span (*Afrane et al., 2006*), partly due to faster blood-meal digestion and frequent blood-feeding (*Afrane, Githeko & Yan, 2012*). Based the climate model projection of (*Lindsay & Dahlman, 2021*), a 0.6 °C increase in environmental temperature during the previous four decades since 1972 (historical data of (*Ismail, Notananda & Schepens, 1975*;

*Ismail, Phinichpongse & Boonrasri, 1978*) would have implications for malaria elimination programmes and future vulnerable populations (*Bureau of Vector Borne Disease DoDC, 2019*), as people move into areas more suitable for transmission risk in Thailand and the GMS (*Sudathip et al., 2021*). Given the limitation of this evaluation, we suggest a future study comparing paired HDNT and HLC methods in the SFS and an outdoor setting during the peak season of high abundance of the *An. minimus* complex, using a Latin square.

## CONCLUSION

This study did not provide support for a relationship between HDNT and HLC in an SFS, presumably due to differences in environmental conditions and the physical barriers associated with the HDNT method. Further studies are required to investigate and compare the sampling efficacy of HLC and other sensitive, cost-efficient, exposure-free, surveillance tools to estimate the rate humans are bitten by mosquitoes.

## ACKNOWLEDGEMENTS

The authors would like to express their gratitude to the study volunteers who tirelessly devoted their time and effort to this study and the Medical Entomology Laboratory, Department of Entomology, Faculty of Agriculture, Kasetsart University for assistance with this research.

### Funding

This research was funded by the Kasetsart University Research and Development Institute (KURDI), Bangkok, Thailand (Grant # FF (KU) 14.64) and the Office of the Ministry of Higher Education, Science, Research and Innovation; and the Thailand Science Research and Innovation through the Kasetsart University Reinventing University Program 2021. The funders had no role in study design, data collection and analysis, decision to publish, or preparation of the manuscript.

### Grant Disclosures

The following grant information was disclosed by the authors:
Kasetsart University Research and Development Institute (KURDI), Bangkok, Thailand: # FF (KU) 14.64.
The Office of the Ministry of Higher Education, Science, Research and Innovation; and the Thailand Science Research and Innovation through the Kasetsart University Reinventing University Program 2021.

### Competing Interests

The authors declare there are no competing interests.

## Author Contributions

- Chanly Yan conceived and designed the experiments, performed the experiments, analyzed the data, prepared figures and/or tables, authored or reviewed drafts of the article, and approved the final draft.
- Jeffrey Hii conceived and designed the experiments, analyzed the data, prepared figures and/or tables, authored or reviewed drafts of the article, and approved the final draft.
- Ratchadawan Ngoen-Klan analyzed the data, prepared figures and/or tables, authored or reviewed drafts of the article, and approved the final draft.
- Manop Saeung performed the experiments, authored or reviewed drafts of the article, and approved the final draft.
- Theeraphap Chareonviriyaphap conceived and designed the experiments, authored or reviewed drafts of the article, and approved the final draft.

## Human Ethics

The following information was supplied relating to ethical approvals (i.e., approving body and any reference numbers):

Ethics approval for the study was provided by the Research Ethics Review Committee for Research Involving Human Research Participants, Kasetsart university (Certificate of approval No. CAO63/035). Formal ethical clearance of study protocol and volunteer collector informed consent was obtained before commencing trials. For the HLC method, assisted guidance was provided to ensure probing mosquitoes were collected prior to biting, including information, awareness of the risks of wild mosquitoes in pathogen transmission, and guarantee of medical care during the duration of the study.

## Data Availability

The raw data are available in the Supplementary Files.

## Supplemental Information

Supplemental information for this article can be found online at http://dx.doi.org/10.7717/peerj.13865#supplemental-information.

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
