# Peer review of "Semi-field evaluation of human landing catches versus human double net trap for estimating human biting rate of Anopheles minimus and Anopheles harrisoni in Thailand"

_PeerJ, doi:10.7717/peerj.13865_

## Round 0.1 · original submission · Major Revisions

The review process is now complete, and three thorough reviews from qualified referees are included at the bottom of this letter. Reviewers and myself, agree the manuscript presents an original and pertinent goal that deserves to be published. Although there is considerable merit in your paper, we identified some concerns that must be considered in your resubmission. Please, reorganize the manuscript's structure to make it clear and fluent to the readers. The quality of figures and data presentation need to be improved. Please, correct all of the inconsistences in the Results section as recommended.

Reviewer 1 ·

Basic reporting

Nothing to add

Experimental design

This must be improved. It should be stated clearly that the lab bred mosquitoes and wild mosquitoes were released on different days. 2nd Why were 50 mosquitoes released near the HLC collection and 50 near the HDNT. Why were the 100 mosquitoes not released in the centre? Please explain

Validity of the findings

Nothing to add

Additional comments

No further comments

Reviewer 2 ·

Basic reporting

The manuscript entitled “A semi-field evaluation of the use of human landing catches (HLC) versus human double net trap (HDNT) for estimating reliable human biting rate of Anopheles minimus (Diptera: Culicidae) in Thailand” deals with a very important aspect of the malaria surveillance system. The search for a good and available exposure-free surveillance tool for substituting HLC is still a great challenge to public health worldwide. I have the following comments to improve the quality of the manuscript:
- Although generally well-written, the manuscript would benefit from a thorough proof-reading for English grammar.
- The quality of the figures is not adequate, I suggest improving them.
- Lines 82: Please, add the meaning of the abbreviations ITN.
- Lines 91-93: Please, add the HDTN reference at the end of the sentence.

Experimental design

- Lines 155-156, 167: In both methods, was it the same person who collected the 12 hours in a row?
- In lines 175-176 it was stated that “The collectors wore shorts up to the knee, sandals, and a long-sleeve shirt and refrained from smoking, alcohol consumption and washing with soap.”, however, in Fig 1B the collector was wearing a short-sleeved red shirt.
- Some information from the “Mosquito trap collections” section is repeated in “Experimental design”. I suggest that the information contained in the sentences of lines 155-157 and 164-168 are only described in the "experimental design" section. I also suggest moving the sentence in lines 175-176 to the first paragraph, about HLC, of the “Mosquito trap collections” section.
- Fig 1 legend: I suggest that the letters in the figure that represent the four compartments of the SFS are lowercase and that this information is included in the legend.
- Were wild mosquitoes analyzed to see if they were infected?
- Please clarify why mosquitoes resting on the walls of the SFS rooms were included in the analysis, as they did not respond/were attracted to the traps (HLC and HDTN), and the main objective of the study was to evaluate the efficacy of an alternative method compared to HLC.

Validity of the findings

Overall the results are presented in a clear and succinct way using well designed figures and tables. However, I strongly recommend improving the quality of the figures. In addition, figure legends are too small and should be made clearer.
Please verify the lines 283-285, in which the sentence is incomplete, as well in lines 331-332.
The discussion is good and gives a clear understanding of the challenges in malaria vectors surveillance and the need for new tools. However, the sentence in lines 407-414 is difficult to understand and should be rewritten, perhaps divided in two sentences.

Reviewer 3 ·

Basic reporting

This manuscript on the comparison of HDNT and HLC presents an interesting and worthwhile work when many countries are now prohibiting collecting on humans (HLC). Alternative collecting methods are then in great need. However, this study is not easy to follow, the manuscript is dense with lots of results showed in 4 figures and 5 tables. The text needs to be better structured to make its reading more fluent. For instance, in the results section, the authors decided to present their results for An. harrisoni first, then followed by An. minimus. However, subsections would help to better understand what is being analyzed for each species. For instance, as it is for An. harrisoni, subsection 1 would be on Recapture rates, subsection 2 on Resting rates, subsection 3 on Quarterly collection rates, etc. The same format should be kept for both species, which is not the case because for An. minimus it starts with landing rates, recapture rates, resting, etc. In addition, more references to the tables should be made into the text in order to understand the values listed in the manuscript. On lines 265, 267, 304 and 308, refer to Table 4; on line 302 to Table 3.
To ease the reading of the manuscript, the quarterly night section should also be written in sequence, starting with 18:00-21:00 h and ending with the 3:00-6:00 quarter, which is not the case in the paragraph on An. harrisoni (lines 275-278).

Experimental design

A major comment concerns the fact this study was done on wild An. harrisoni collected in Kanchanaburi Province, while An. minimus is coming from the laboratory, a population that has been kept in an insectarium since 1993. Could you comment on the fact some results may have been impacted by this difference of origins in which An. minimus may have lost its wild behavior? Why haven't you used wild An. minimus (from another study sites) such as An. harrisoni?
It is not clear how many An. harrisoni specimens were species-identified using the AS-PCR assay and then used in the experiments. How sure are you that the specimens used in the experiments belonged to An. harrisoni?

Validity of the findings

Several errors or inconsistencies between values mentioned in the manuscript and tables have been found. Check on line 260, p=0.323 in the text but p=0.202 in Table 2; on line 267, value is 633 in the text while it’s 804 in Table 4; on line 278, the value is 6.36 per night in the text and 5.72 in Table 5; on line 302, it’s 37.87% in the text and 37.4% in Table 3.
Calculations should be double-checked to be sure they are correct because errors may change the P-values, but also the interpretation of the results and the conclusions of the study.

Additional comments

Many minor comments and corrections as follow:
- The title is too long and you don’t refer to both species of the minimus complex. You may change it for “A semi-field evaluation of human landing catches (HLC) versus human double net trap (HDNT) for estimating human biting rate of Anopheles minimus and An. harrisoni in Thailand”.
- The corresponding author seems to be missing from the list of authors on the manuscript to be reviewed.
- An. minimus is a lab strain; therefore, it was identified by PCR and there is no doubt on the species. Then, no need to write An. minimus s.s., but instead you should write An. minimus because it goes without saying that you worked at the species level.
- Update the Background section with values on malaria provided by WHO 2021, World Malaria report with 2020 data.
- Malaria foci are mainly distributed along international border. Add “international” on line 49.
- Write “former” species A, former species C and former species E after each species of Minimus Complex (line 58).
- After Greater Mekong Subregion add “(GMS)” (lines 70-71), as GMS is mentioned later on line 440.
- On lines 72-73, write “Differences in responses … exposure rates of species or … of the Minimus complex.” Next lines (74-75), use the passed with “showed”.
- Be consistent and write all along the manuscript HDNT instead of HDN (lines 97-98, 100, 105, 256, Fig. 2, etc).
- Write “2021” on line 170, instead of 20221.
- Lines 257-259, write “Significant higher abundance of 85% … of laboratory strain (An. minimus) were collected in …
- On lines 264 and 266, precise the species and replace Anopheles by An. harrisoni.
- On line 303, add a verb in the sentence.
- On line 351, what do you mean by “errors of mosquito collection”?
- On line 360, write “anthropophily” instead of anthropophagy.
- On line 368, precise “greater numbers collected in HDNT compared to what, HLC?
- On line 393, write “albopictus” instead of Albopictus.
- On line 416, the term of “variable-s” is written twice, replace one of them to avoid repetition.
- In the sentence (lines 417-419) there is no verb. Add one.
- On line 421, add “Thailand” after Saraburi Province.
- The sentence at the end of the discussion (lines 438-439) should be rephrased as the part “would have implication for malaria elimination target” is not clear.
- All “Anopheles” and “versus” terms must be written in italics in the manuscript, many aren’t. The same for all Latin names (species).
- For the bibliography, 3 articles are missing, such as WHO (2021) in the Background section on lines 42-47, Carnevale & Manguin (2021) on lines 67-69; Garros et al (2006) on line 358.
- Several references are also incomplete including Durnez & Coosemans (2013), Garros et al (2005), Ismail et al (1978), MoPH (2019), Somboon (1993).
- Write properly “An. sawadwongporni” on line 492.
- Species must be written in italics such as Anopheles minimus (lines 528-529), An. dirus (line 573).
- Finally, the journals must be written following the format given by PeerJ. Some are written in full while others are abbreviated. Be consistent and follow the journal guidelines.
- Figure 2b: change the title as “Mosquitoes was predicted for volunteers” doesn’t mean anything.
- Figure 4: why is there a mixture of An. minimus (d, h) and An. minimus s.s. (c, g)? This inconsistency raises question on the identification of the species.
- Table 1: spell out abbreviations (W, L, HLC, HDNT). Anopheles in italics.

---

## Round 0.2 · accepted · Accept

The authors addressed all points raised by the reviewers.

Reviewer 1 ·

Basic reporting

No comments

Experimental design

No comments

Validity of the findings

No comments

Additional comments

Can be accepted

Reviewer 2 ·

Basic reporting

All comments have been addressed.

Experimental design

All comments have been addressed.

Validity of the findings

All comments have been addressed.

Additional comments

Thank you for addressing all my comments. I have read your revision through and I hope I have contributed for a better communication.